# Angiotensin II Type I Receptor (AT1R): The Gate towards COVID-19-Associated Diseases

**DOI:** 10.3390/molecules27072048

**Published:** 2022-03-22

**Authors:** George El-Arif, Shaymaa Khazaal, Antonella Farhat, Julien Harb, Cédric Annweiler, Yingliang Wu, Zhijian Cao, Hervé Kovacic, Ziad Abi Khattar, Ziad Fajloun, Jean-Marc Sabatier

**Affiliations:** 1Faculty of Sciences 2, Department of Biology, Lebanese University, Campus Fanar, Jdeidet El-Matn, Beirut P.O. Box 90656, Lebanon; george.elarif@ul.edu.lb (G.E.-A.); antonella_farhat@ul.edu.lb (A.F.); 2Faculty of Sciences 3, Department of Biology, Lebanese University, Campus Michel Slayman Ras Maska, Tripoli 1352, Lebanon; shaymaa.khazaal@ul.edu.lb; 3Faculty of Medicine and Medical Sciences, University of Balamand, Dekouene Campus, Sin El Fil P.O. Box 55251, Lebanon; julienharb0408@gmail.com; 4Research Center on Autonomy and Longevity, Department of Geriatric Medicine and Memory Clinic, 44312 Angers, France; cedric.annweiler@chu-angers.fr; 5Laboratoire de Psychologie des Pays de la Loire, LPPL EA 4638, SFR Confluences, University of Angers, 44312 Angers, France; 6State Key Laboratory of Virology, Modern Virology Research Center, College of Life Sciences, Wuhan University, Wuhan 430072, China; ylwu@whu.edu.cn (Y.W.); zjcao@whu.edu.cn (Z.C.); 7Institut de Neurophysiopathologie (INP), Aix-Marseille Université CNRS, 13385 Marseille, France; herve.kovacic@univ-amu.fr; 8Laboratory of Georesources, Geosciences and Environment (L2GE), Microbiology/Tox-Ecotoxicology Team, Faculty of Sciences 2, Lebanese University, Campus Fanar, Jdeidet El-Matn, Beirut P.O. Box 90656, Lebanon; 9Laboratory of Applied Biotechnology (LBA3B), Azm Center for Research in Biotechnology and Its Applications, EDST, Lebanese University, Tripoli 1300, Lebanon

**Keywords:** SARS-CoV-2, COVID-19, Ang II–AT1R axis, ACE2, AT1R downstream signaling pathways, multiple system damages, ARBs

## Abstract

The binding of the severe acute respiratory syndrome coronavirus 2 (SARS-CoV-2) spike glycoprotein to its cellular receptor, the angiotensin-converting enzyme 2 (ACE2), causes its downregulation, which subsequently leads to the dysregulation of the renin–angiotensin system (RAS) in favor of the ACE–angiotensin II (Ang II)–angiotensin II type I receptor (AT1R) axis. AT1R has a major role in RAS by being involved in several physiological events including blood pressure control and electrolyte balance. Following SARS-CoV-2 infection, pathogenic episodes generated by the vasoconstriction, proinflammatory, profibrotic, and prooxidative consequences of the Ang II–AT1R axis activation are accompanied by a hyperinflammatory state (cytokine storm) and an acute respiratory distress syndrome (ARDS). AT1R, a member of the G protein-coupled receptor (GPCR) family, modulates Ang II deleterious effects through the activation of multiple downstream signaling pathways, among which are MAP kinases (ERK 1/2, JNK, p38MAPK), receptor tyrosine kinases (PDGF, EGFR, insulin receptor), and nonreceptor tyrosine kinases (Src, JAK/STAT, focal adhesion kinase (FAK)), and nicotinamide adenine dinucleotide phosphate (NADPH) oxidase. COVID-19 is well known for generating respiratory symptoms, but because ACE2 is expressed in various body tissues, several extrapulmonary pathologies are also manifested, including neurologic disorders, vasculature and myocardial complications, kidney injury, gastrointestinal symptoms, hepatic injury, hyperglycemia, and dermatologic complications. Therefore, the development of drugs based on RAS blockers, such as angiotensin II receptor blockers (ARBs), that inhibit the damaging axis of the RAS cascade may become one of the most promising approaches for the treatment of COVID-19 in the near future. We herein review the general features of AT1R, with a special focus on the receptor-mediated activation of the different downstream signaling pathways leading to specific cellular responses. In addition, we provide the latest insights into the roles of AT1R in COVID-19 outcomes in different systems of the human body, as well as the role of ARBs as tentative pharmacological agents to treat COVID-19.

## 1. Introduction

In late autumn 2019, the global health and economic management were dominated by a novel virus, reported from the city of Wuhan (China), designated as the severe acute respiratory syndrome coronavirus 2 (SARS-CoV-2) [1], which is the causal agent of the disease named COVID-19 [2]. The SARS-CoV-2 virions are spherical in shape, with a diameter of approximately 60–140 nm, and have a nucleocapsid core enclosed by a lipid bilayer envelope. The nucleocapsid core contains the viral genome, a single-stranded, non-segmented positive-sense RNA, which has a 5′ cap and a 3′ poly-A tail, with a length of ~26.4 to ~31.7 kb [1]. The lipid bilayer envelope has structural glycoproteins, including the spike (S) protein, the membrane (M) protein, the envelope (E) protein, and the hemagglutinin-esterase (HE) protein. The latter binds to the terminal sialic acid residues on the host cell membrane glycoproteins and manifests an acetyl-esterase activity [3]. The nucleotide sequence analysis of the viral RNA genome has been shown to be closely correlated to that of the beta-coronavirus responsible for the epidemic of SARS in the Asia Pacific (and Canada) in 2002–2003 [4]. The acute respiratory disease caused by SARS-CoV-2 can range from an asymptomatic form to a severe acute respiratory distress syndrome (ARDS). Multiorgan dysfunction and failure are thus common as a result of increased vascular permeability, inflammation, and blood clot formation (coagulation), which eventually leads to mortality [5]. Until now, five SARS-CoV-2 variants have been identified, and the latest variant (B.1.1.529) was designated as omicron by the world health organization (WHO) on 26 November 2021. This variant has cumulated several mutations that may have an impact on its behavior [6].

The membrane-bound angiotensin-converting enzyme 2 (ACE2), a significant component of the renin–angiotensin system (RAS), is the human site of SARS-CoV-2 entry into host cells [5,7]. ACE2 is a monocarboxypeptidase expressed on the surface of a wide variety of different cell types, including epithelial cells lining the respiratory tract, cardiac fibroblasts, cardiomyocytes, endothelial cells, vascular smooth muscle cells (VSMCs), kidneys, intestine, and in the central nervous system (CNS) [5,8]. After the S-protein binding to ACE2, a membrane protease (TMPRSS2) alters the S-protein, allowing the cell membrane to fuse with the viral envelope. Therefore, this will be reflected by the downregulation of ACE2 following the viral infection [5,9]. Once in the cytoplasm, the viral genome codes for the synthesis of proteins necessary for the replication and spread of SARS-CoV-2 [9].

The RAS is a complex and dynamic two-axis-molecular cascade present in different organ systems, which has an essential function in neural, pulmonary, renal, cardiovascular, and immune homeostasis [10]. The classical axis comprises the ACE, angiotensin II (Ang II), and the angiotensin type I receptor (AT1R) [10]. This system mediates, besides the body fluid homeostasis, several deleterious effects through Ang II–AT1R signaling such as vasoconstriction, fibrosis, inflammation, cellular growth, migration, cardiac hypertrophy, thrombosis, and reactive oxygen species (ROS) production [2,10]. A counter regulatory axis is formed by ACE2, Ang-(1–7), and the Mas receptor (MasR) [10]. Ang-(1–7) holds anti-inflammatory, antifibrotic, antiproliferative, antioxidative, vasodilator, and antithrombotic properties exerted mainly via the MasR [2]. Moreover, ACE2 degrades Ang II to Ang-(1–7), thereby reducing Ang II effects on vasoconstriction, sodium retention, and fibrosis [11]. As a result, decreased ACE2 levels following SARS-CoV-2 infection may lead to (i) Ang II upregulation, resulting in classical RAS axis overactivity and (ii) Ang-(1–7) depletion, reducing the activity of the protective alternative RAS axis [5,10] (Figure 1).

The physiological effects of Ang II are mediated mainly by AT1R and Ang II type 2 receptor (AT2R) [12,13,14]. AT1R effects include control of blood pressure, contraction of blood vessels, electrolyte balance, aldosterone synthesis and release from the adrenal cortex [15,16]. Furthermore, each receptor transmits opposing effects, with AT1R mediating vasoconstriction, proliferation, fibrosis, and inflammation and AT2R mediating vasodilation, antifibrosis, and anti-inflammation (Figure 1) [12,14]. AT1R and AT2R are G protein-coupled receptors (GPCRs) sharing a sequence identity of ~30% but having the same affinity for Ang II, which is their main ligand [14,17]. It has been shown that Ang II binding to the AT1R is involved in the pathophysiology of many different tissues. In fact, extreme Ang II–AT1R signaling induces the promotion of vascular remodeling and initiates the progression of atherosclerosis by producing endothelial dysfunction. Activation of AT1Rs in cardiomyocytes generates cellular hypertrophy, while binding of Ang II to AT1Rs on the surface of cardiac fibroblasts results in cardiac fibrosis by stimulating the synthesis of extracellular matrix proteins [18]. Moreover, findings demonstrated that activation of AT1R promotes the release of Ang II induced ROS, inflammation, and angiogenesis via the activation of several signaling pathways including the nicotinamide adenine dinucleotide phosphate (NADPH) oxidase (NOX), nuclear factor κ-light-chain-enhancer of activated B cells (NF-κB), extracellular signal-regulated kinases, (ERK1/2), mitogen-activated protein kinase (MAPK), and signal transducer and activator of transcription 1 (STAT1) pathways [19,20]. Unbalanced AT1R signaling in the lungs is associated with airway inflammation, bronchial hyper-responsiveness, fibrosis, and pulmonary hypertension [21,22].

The COVID-19 pandemic has prompted researchers to look for viable treatments, including vaccinations, antiviral medications that target the SARS-CoV-2 virus, and methods to manage disease pathology and complications caused by interactions with host components. Presently, 46.6% of the world population has been fully vaccinated [6]. However, it has been also shown that the treatment with RAS component inhibitors may have positive effects in treating COVID-19 diseases [23]. The most commonly used drugs include renin inhibitors, ACE inhibitors (ACEIs), and AT1R blockers (ARBs) [13]. ACEIs and ARBs have been shown to improve pulmonary capillary permeability, reduce apoptosis, and attenuate acute lung injury following SARS-CoV-2 infection [23]. In addition, ARBs might attenuate ARDS and cytokine storms in COVID-19 [2]. ARBs are nonpeptide antagonists, which include antihypertensive drugs such as losartan, olmesartan, candesartan, telmisartan, valsartan, irbesartan, eprosartan and azilsartan. These ARBs are now extensively used for treating cardiovascular diseases (CVDs), including heart failure, hypertension, cardiac hypertrophy, and arrhythmia [13]. ARBs block the binding of Ang II to the AT1R, thus making it available for the AT2R by turning off the vasoconstrictive, proliferative, and inflammatory effects [24]. Therefore, the significant clinical effectiveness of RAS blockers, together with their benefits for target organs, provides evidence of the continued clinical importance of RAS-targeted drug development [18].

On that account, the present review article focuses on the distribution and function of AT1 receptors, with the aim to help determine the pathways involved in their function. Certainly, their activation in response to SARS-CoV-2 infection, as well as their negative consequences on several human body systems, is emphasized. Additionally, the AT1R blockers and their efficacy in the treatment of COVID-19 are discussed in this review.

## 2. Angiotensin II Type I Receptor (AT1R) and COVID-19

### 2.1. The AT1R: Structural Characteristics, Polymorphisms, Localization, Activation, and Signaling Pathways

#### 2.1.1. Structural Characteristics

Until now, four types of Ang II receptors have been identified (AT1R, AT2R, AT3R, and AT4R) among which the AT1R is the most well studied [25]. In fact, most Ang II physiological effects are mediated through AT1R [14]. AT1R is a GPCR that requires Ang II binding for its activation [14,15,25]. Therefore, it belongs to the canonical seven-transmembrane α-helical superfamily, with an intracellular C-terminal domain, an extracellular N-terminal domain, three intracellular loops (ICL1, ICL 2, ICL3), an amphipathic helix VIII, and three extracellular loops (ECL1, ECL2, ECL3) [13,14,26]. The amino acid sequence of AT1R protein, with a molecular weight of 41 kDa, consists of 359 residues [14,26], sharing 20–35% homology with other GPCRs [26].

The extracellular part of AT1R consists of the N-terminal domain, ECL1 (Glu91-Phe96) linking helices II and III, ECL2 (His166 to Ile191) linking helices IV and V, and ECL3 (Ile270 to Cys274) linking helices VI and VII [13]. This extracellular domain is characterized by three consensus N-glycosylation sites (Asn4 within the NH2-terminus and Asn176 and Asn188 in the ECL2) [14,26]. Mutation of these sites has been determined to have no effect on agonist binding [14]. Among many residues found on the extracellular part of the receptor, four cysteine residues form two disulfide bridges and are essential for Ang II binding [14,26]. The first bridge (Cys101–Cys180) connecting the ECL2 and helix III is highly conserved in other GPCRs, while the second disulfide bridge (Cys18-Cys274) connects the N-terminus to the ECL3 [13,26]. The ECL2 of AT1R exhibits a β-hairpin secondary structure, a common motif among GPCR peptides, which serves as an epitope for the harmful agonistic autoantibodies in malignant hypertension [13].

The intracellular part of AT1R contains ICL1 (Lys58 to Val62) linking helices I and II, ICL2 (Val131 to Arg137) linking helices III and IV, ICL3 (Leu222 to Asn235) linking helices V and VI, and the C-terminal helix VIII [13]. Similar to other GPCRs, the conserved NPxxY motif in helix VII and the D(E)RY motif in helix III at the intracellular ends of the AT1R transmembrane domain, were believed to be involved in receptor activation. The C-terminal helix VIII is also important for receptor internalization and coupling to G protein activation and signaling. This helix is angled away from the membrane, unlike the other GPCRs, which have helixes that run parallel to the membrane [13]. The carboxyl-terminal cytoplasmic tail is rich in serine/threonine and tyrosine residues, with three putative phosphorylation sites by G protein receptor kinases (protein kinase C “PKC”) [14,26]. Modifications of these sites may alter receptor function in CVDs [14].

Ang II binds to a specific binding domain of AT1R. In fact, the carboxyl group of Phe8 in Ang II binds to Lys199 in the transmembrane domain (TM5) of the AT1R. The Ang II binding to AT1R is further stabilized by two other interactions; the first between His183 in ECL2 and Asp1 in Ang II, and the second between Asp281 in TM7 and Arg2 in Ang II. Moreover, interactions of His256, Val254, and Phe259 in TM6 with the aromatic side chain of Phe8 of Ang II, and interaction of Asn111 in TM3 with Tyr4 of Ang II have been postulated to contribute to Ang II docking with the receptor [26]. As a result, it is plausible that pharmacological targeting or mutagenesis of these locations could block Ang II–AT1R binding, thereby preventing severe consequences of COVID-19 when the Ang II–AT1R axis is activated.

Although AT1R is typically activated by Ang II, additional ligands such as AT1R autoantibodies and mechanical stress have been reported to activate this receptor in a variety of clinical conditions such as hypertension, pre-eclampsia, and cardiac hypertrophy [27].

#### 2.1.2. AT1R Polymorphisms and Associated Effects

Polymorphisms in GPCRs were shown to be associated with different phenotypes and to predispose to certain types of diseases [15]. More than 700 mutations, that can overactivate or inactivate the receptors, have been identified and linked to more than 30 distinct human illnesses [15]. Minor information about AT1R polymorphisms in SARS-CoV-2 patients is found, despite their importance as risk factors in COVID-19 such as those implicated in diabetes, hypertension, and cardiovascular diseases (CVDs) [28].

The AT1R gene is polymorphic, and several variants have been discovered. The human AT1R gene is located on chromosome 3 (3q21–q25) and is constituted of four introns and five exons, with a >55 kb length [14,16,26,29,30,31,32]. The first four exons constitute the 5′UTR region whereas the last exon contains the coding region [30,32]. Among the five frequent SNPs in the AT1R gene (G1517T, A1166C, and A1878G located in the 3′UTR, T573C, and A1062G located in the coding region) [16,30], A1166C (ID: rs5186) is the most studied [16,29,33].

The study of Izmailova et al. on the Ukrainian population showed an association between COVID-19 severity and the A1166C polymorphism. When compared with A-allele carriers, C allele carriers had significant COVID-19 pathogenicity, as well as higher oxygen requirement due to cardiovascular problems [28]. In fact, the C allele has been shown to be associated with many CVDs such as systolic blood pressure, left ventricular hypertrophy, hypertension, aortic stiffness, myocardial infarction, carotid intimal–medial thickening, coronary artery disease, and stroke [32,33].

Amir et al. examined the AT1R A1166C polymorphism in patients with systolic heart failure and its relation to clinical outcomes. It was shown that the CC genotype was associated with more advanced disease and severe abnormalities of the renal function (*p* = 0.008), as well as a significantly higher mortality rate (*p* = 0.008). Increased AT1R expression and receptor activity were seen in homozygous patients, which could explain the RAS’s amplified neurohormonal activation [34].

#### 2.1.3. AT1R Localization, Activation, and Signaling Pathways

AT1R is expressed in different organs including heart, skin, kidneys, blood vessels, skeletal muscles, brain, liver, lungs, and adrenal glands [14,25,29]. Almost all physiological and pathophysiological effects of Ang II are mediated through the AT1R [13]. AT1R induces vasoconstriction, proliferation, inflammation, and fibrosis, which are related to CVDs [15,31]. In the brain, through AT1R, Ang II controls fluid homeostasis and autonomic pathways regulating the cardiovascular and neuroendocrine systems [25]. The AT1R regulation can provide a link between hypertension and various disorders such as hyperlipidemia and hyperinsulinemia. Low-density lipoproteins (LDLs) have been shown to upregulate AT1R via post-transcriptional mRNA stabilization. Insulin has been found to have the same effect, which provides molecular evidence for an association between hyperinsulinemia and hypertension [14].

Activation of the AT1R by Ang II leads to G protein-mediated signaling pathways in a tissue-specific manner. Signal transduction occurs via a series of intracellular second messengers, including phospholipases, calcium signaling, and PKC [15,32] (Figure 2). After AT1R activation, GPCR-kinases promote the phosphorylation of the receptor and β-arrestins recruitment, leading to receptor internalization and activation of other signaling pathways [14,35]. AT1R activation by Ang II triggers the activation of phospholipase C (PLC), hydrolysis of membrane phospholipids, and thus the liberation of diacylglycerol (DAG) (which activates PKC) and inositol trisphosphate (IP3) (which mobilizes intracellular calcium). These signals have been linked to the classical effects of Ang II characterized by vasoconstriction, aldosterone release, and water and salt imbalance [35]. Besides the activation of direct effectors, AT1R signaling also promotes downstream phosphorylation of different kinases and substrates that are involved in alterations of extracellular matrix, gap junction formation, and ion channel functionality [15]. AT1R cross-talks with several tyrosine kinases: receptor tyrosine kinases (epidermal growth factor “EGFR”, platelet-derived growth factor “PDGF”, insulin receptor) and nonreceptor tyrosine kinases (c-Src family kinases, Ca^2+^ dependent proline-rich tyrosine kinase 2 “Pyk2”, focal adhesion kinase “FAK”, and Janus kinases “JAK”) [14]. The capacity of AT1R to activate tyrosine kinase-related pathways explains the effect of Ang II in vasculature growth and remodeling that occurs during metabolic disturbances and CVDs [35]. JAK/STAT signaling is part of an autocrine loop for Ang II generation, which reinforces its action on cardiomyocytes to elicit cardiac hypertrophy [25]. In fact, this pathway upregulates angiotensinogen formation, resulting in elevated Ang II production [25]. The JAK of the activated JAK/STAT signaling pathway transduces the activity of extracellular ligands, including growth factors and cytokines. The binding of the latter to cell surface receptors leads to an intracellular activation of JAK proteins, which phosphorylate the receptor tyrosine residues for the recruitment of STAT proteins to the receptors, where they are also tyrosine-phosphorylated by JAKs. Phosphorylated STATs form homo- or heterodimers and then translocate to the cell nucleus, where they induce transcription of target genes. JAK/STAT signaling regulates various genes in immune effector cells (IECs) and T lymphocytes [36]. In addition, many of Ang II pathologic effects in the vasculature occur via activation of NADH/NADPH oxidases and generation of reactive oxygen species (ROS) [14]. The NADPH oxidases catalytic subunits (NOX family) are consisted of seven members: Nox1- Nox5, and dual oxidases (Duox1 and Duox2). Upon activation, Nox1, Nox2, and Nox5 mainly produce superoxide, whereas Nox4 mainly produces H_2_O_2_ [18]. AT1R also activates serine/threonine kinases such as PKC and mitogen-activated protein kinase “MAPKs” (including extracellular signal-regulated kinases “ERK1/2”, p38-MAPK, and c-Jun NH2-terminal kinase “JNK”) that are implicated in cell growth and hypertrophy [14]. Furthermore, AT1R stimulates NF-κB pathway, which is involved in the development of oxidative stress and the activation of proinflammatory cytokines [25] (Figure 2).

The activation of the above-mentioned pathways is known to be time-dependent. For example, activation of the G protein-dependent pathway and generation of IP3 and DAG occur in seconds, while MAP kinase and JAK/STAT activations occur in minutes to hours after the initial activation of AT1R [14]. Furthermore, differences in receptor–ligand affinity, alteration in trafficking patterns, AT1R structural modifications, and the local tissue environment all appear to contribute to the ultimate effects of Ang II signaling [14].

### 2.2. AT1R Role in COVID-19 Diseases

In SARS-CoV-2-induced pathogenesis, the virus infects host cells through the ACE2 receptor on the respiratory epithelium. The viral invasion and downregulation of the ACE2 site promotes an increase in Ang II serum levels and, as a result, AT1R overstimulation [24]. Hospitalized patients showed a wide spectrum of manifestations including pulmonary damage such as pneumonia, airways severe damage, pulmonary edema, and ARDS, the causative mechanism of death in 70% of fatal COVID-19 cases [37]. Furthermore, the aberrant activation of AT1R by hyperacute excess of Ang II reflects the mechanism of cytokine storm in COVID-19, which causes lung inflammation by releasing proinflammatory cytokines that have proliferative, thrombotic, and tissue-destructive effects [2]. The SARS-CoV-2 infection has been shown to affect systems other than the respiratory tract, including the cardiovascular, renal, nervous, and gastrointestinal (GI) systems [38]. This could be explained by the diverse expression of ACE2 receptor and TMPRSS2 in nasal goblet secretory cells, lung alveolar epithelial type II cells, colonocytes, cholangiocytes, pancreatic β-cells, GI epithelial cells, and renal proximal tubules and podocytes [38]. All of these exaggerated reactions in COVID-19 patients can lead to multiorgan failure and death [5,39] (Figure 3).

#### 2.2.1. AT1R and the Immune System

The induced immune responses to SARS-CoV-2 infection have two distinct arms: (1) protective, in which the adaptive immune response plays an important role in eradicating the virus early in the disease and (2) damaging, in which immune responses can cause tissue damage by over-releasing proinflammatory cytokines [5]. The severe and fatal form of COVID-19 is marked by the cytokine storm syndrome (CSS), which is caused by a dysregulated immune response characterized by excessive activation of innate immunity, an inhibition of interferon signaling by the virus, T-cell lymphodepletion, activation of neutrophils and monocyte-derived macrophages as mediators of hyperinflammation, and the excessive and uncontrolled production of proinflammatory cytokines (particularly IL-6, IL-1 and tumor necrosis factor TNF-α) [38], and chemokines [37]. CSS could thus be defined by a massive overproduction of cytokines and chemokines in response to exponential viral replication, which could result in endothelial and epithelial cell damage, followed by vascular leakage, eventually leading to ARDS, multiorgan failure, and death [40]. The cytokine storm begins from a local site of inflammation and then spreads throughout the entire body [39]. This could explain the hematological findings in COVID-19 severe cases including leukocytosis, lymphopenia, thrombocytopenia, neutrophilia, elevations in erythrocyte sedimentation rate, C-reactive protein (CRP), IL-6, fibrinogen, ferritin, lactate dehydrogenase, and D-dimer [38]. Moreover, macrophages have been shown to be involved in CSS. Normally, M1 macrophages are important for inducing immune responses against viral infection, which could be impaired by SARS-CoV-2 infection. Following the activation of Toll-like receptors (TLRs) and pattern recognition receptors (PRRs), alveolar M1 macrophages are activated in large amounts, causing macrophage activation syndrome (MAS), which is involved in SARS-CoV-2-induced ARDS [5] (Figure 4). The inflammatory response is important in COVID-19, and an inflammatory cytokine storm worsens the disease severity [41].

It is well known that these features are induced by the dysregulated RAS. Ang II has been shown to play a role in regulating the immune system (Figure 4). In fact, AT1R is found on the surface of a variety of immune cells, including T cells, which possess endogenous RAS, express functional RAS components, and locally produce Ang II [42,43,44]. The latter promotes immune cells’ proliferation, migration, differentiation, adhesion, and effector function, and acts as a costimulatory molecule important for T-cell activation. These effects, however, are induced by AT1R binding, which is followed by increased production of proinflammatory cytokines by CD4+ T cells and perforin by CD8+ T cells, as well as increased capacity to adhere and migrate via upregulation of adhesion molecules and chemokine receptors [42].

The Ang II–AT1R axis activates the NF-κB/IL-6/STAT3 pathway, which can trigger the amplification of IL-6 via a positive feedback loop of NF-κB and STAT3 coactivation [5,37]. The NF-κB pathway is shown to be involved in the pathogenesis of COVID-19 patients, as well as in the CSS, by inducing the expression of proinflammatory genes, including IL-6, chemokines, and growth factors, critical for the development of various diseases [5,37]. This phenomenon begins with the activation of the metallopeptidase ADAM domain 17 (ADAM17) by the Ang II–AT1R axis via the intracellular p38/MAPK signaling pathway. ADAM17 digests the membrane forms of epidermal growth factor family members and TNF-α generating their soluble forms, all of which stimulate the NF-κB pathway [24,37]. Moreover, ADAM17 operates membrane-bound IL-6Rα to the soluble form (sIL-6Rα) [37]. The sIL-6R–IL-6 complex binds to glycoprotein 130 (gp130), expressed in most cells, and transduces intracellular signaling, followed by the activation of JAK/STAT3 and the induction of excess IL-6 [37,39] (Figure 4). IL-6 amplifier induces chronic inflammatory diseases and the production of proinflammatory cytokines, leading to cytokine storm that characterizes the most severe forms of SARS-CoV-2 infection [5,24,39]. IL-6 is a multifunctional cytokine that transmits cell signaling, regulates immune cells, and possesses a strong proinflammatory effect, with multiple biological functions [41]. It has been shown that patients with IL-6 level > 32.1 pg/mL or CRP level > 41.8 mg/L were more likely to have severe complications [41]. Moreover, IL-6 is known to be one of the strongest predictors of mortality during hospitalization [44]. The increased inflammatory cytokines such as TNF-α and IL-6 induce apoptosis and necrosis in T cells, which are reduced in COVID-19. Since AT1R is also expressed on macrophages and dendritic cells, their activation also leads to the production of inflammatory cytokines (TNF-α, IL-1b, IL-6, and IL-10) and ROS, through NF-κB and AP-1 pathways [44].

It was discovered that inhibiting NF-κB reduced the release of proinflammatory cytokines and chemokines and interfered with their harmful effects, such as multiorgan tissue damage and ARDS [5]. Ang II interacts with both immune cells (neutrophils, mononuclear cells, T cells, and B cells) and nonimmune tissue-resident cells [39]. Interestingly, the activation occurs more easily in nonimmune cells. The activated Ang II synthesis from tissue-resident cells enhances vascular permeability by promoting the production of prostaglandins, vascular endothelial cell growth factor (VEGF), NF-κB, TNFα, IL-1β, and IL-6 [39].

Liu et al. have studied, retrospectively, the ability of IL-6, CRP, and procalcitonin (PCT) to predict mild and severe cases in 140 patients diagnosed with COVID-19. They found that IL-6, CRP, and PCT levels increased by 67.9%, 65.0%, and 5.7%, respectively. The proportion of patients with elevated levels of these proteins in the severe group was significantly higher than in the mild group [41]. Tawinwung et al. have shown that AT1R signaling is essential for T-cell activation and IL-2 production, and the inhibition of this pathway suppressed T-cell activation via an ERK-dependent mechanism [43].

Moreover, Ang II increases ROS production through AT1R-dependent induction of NADPH oxidase (NOX), mostly Nox2 and Nox4 [2], and the phosphatidylinositol-4,5-bisphosphate 3-kinase/protein kinase B (PI3K/Akt) pathway [39]. ROS overproduction may lead to DNA damage and mitochondrial dysfunction (Figure 4). In fact, by opening mitochondrial K-ATP channels and disrupting mitochondrial membrane potential, a positive feedback response is observed, resulting in an increase in mitochondrial ROS (mtROS) production, which triggers the production of proinflammatory cytokines [2]. Furthermore, ROS activates inflammatory responses by inducing redox-sensitive transcriptional factors such as NF-κB and AP-1. Within or near the infected cell, the Ang II-AT1R increases CRP and IL-6 via ROS production. In addition, more inflammatory factors are produced such as TNF-α, MCP-1, tissue factor, plasminogen activator inhibitor-1(PAI-1), which may contribute to the state of overwhelming systemic inflammation and hypercoagulability [40]. Ang II–AT1R has also been shown to exacerbate oxidative stress by disrupting iron homeostasis and increasing labile ferrous iron, as well as ferritin expression in endothelial cells, which could serve as a local cytokine and activate NF-κB via the MAPK pathway. This response results in a rise in inducible NO synthase, by about 100-fold, and IL-1β by 50-fold, with a small increase in intercellular adhesion molecule (ICAM) [2].

Finally, it is noteworthy to mention that obesity is considered a risk factor for COVID-19 severity, and this could be explained by the role of Ang II–AT1R. In fact, it has been shown that human-cultured adipocytes promote the production of IL-6 and IL-8 by the NF-κB-mediated pathway. In addition, it was demonstrated that plasma levels of IL-6 in obese individuals are closely correlated to body mass index (BMI) [2].

#### 2.2.2. AT1R and the Respiratory System

SARS-CoV-2 is known to be transmitted mainly through the respiratory tract by direct or indirect contact [38]. The high expression of ACE2, in the airway epithelial cell types and the alveolar epithelial type II cells in the lung parenchyma, favors viral entry [38]. It has been shown that ACE, Ang II, and AT1R induce lung edema and impair lung function through increased Ang II production [45]. The dysregulated Ang II–AT1R axis activates downstream inflammatory signaling pathways, resulting in increased permeability of pulmonary capillaries, injury to epithelial cells and lung endothelium, and subsequently, acute lung injury [23]. Imai et al. reported that inactivating ACE genetically or treating mice with recombinant human ACE2 (rhuACE2) reduced Ang II levels in the lungs and plasma, as well as edema development and histological alterations, all of which decreased lung injury [45]. In addition, genetic loss of AT1aR in rodents improved significantly lung function and reduced edema formation. Thus, Ang II signaling through its receptor is responsible for ACE2-regulated lung pathology [45].

With all of its proinflammatory, destructive, and profibrotic properties, Ang II–AT1R signaling in the lung increases vascular permeability [2,10] by releasing prostaglandins and vascular endothelial growth factor (VEGF) [2,5]. In fact, when the virus invades the respiratory tract, the vascular endothelial cells and epithelial cells are damaged first, triggering an accumulation of protein-rich edema fluid in the alveoli and lung interstitium, which will activate macrophages and neutrophils to release a large number of inflammatory factors [23]. This axis has also been related to apoptosis in pneumocytes, and subsequent alveolar and bronchial cell death, which contributes to SARS-CoV-2 pathogenesis [2]. The Ang II–AT1R axis activates NF-κB, JAK2/STATs, and MAPK pathways and triggers inflammation [23]. The activation of innate immunity leads to the production of TNF-α, IL-6, IL-8, IL-1β, interferon-inducible protein-10 (IP-10), caspase 3 (CASP3), CASP9, and fibroblast growth factor-7 (FGF-7) [23]. The Ang II–AT1R axis overstimulation promotes endothelial dysfunction directly by Ang II effects and indirectly through immune system activation and hypoxia. Ang II may operate locally in the lung endothelium, by increasing ROS production while decreasing NO production and lung cell death [10,23]. All these pathophysiological mechanisms might result in dyspnea, hypoxia, [10], and severe cases of pneumonia and ARDS [38]. It has been shown that ACE2 loss promotes acute lung injury through AT1R stimulation by Ang II, which could lead to leaky pulmonary blood vessels [45].

ARDS is the most severe form of acute lung injury, with a mortality ranging from 30% to 60% [45]. ARDS is characterized by severe hypoxia, accumulation of inflammatory cells, increased vascular permeability, and pulmonary edema, which could lead to lung fibrosis [45,46]. During ARDS, matrix metalloproteinases are produced, followed by IL-6, TNF-α, and vascular endothelial growth factor production [46]. In fact, Ang II–AT1R induces collagen deposition, oxidative stress, and activation of nucleotide-binding oligomerization, domain-like receptor pyrin domain containing 3 (NLRP3), increasing the release of IL-18 and IFN-γ. The latter activates macrophages, which produce Il-6, IL-8, IL-18, IL-1, and monocyte chemotactic protein 1 (MCP-1), contributing to alveolar epithelial damage. All these features contribute to pulmonary fibrosis and acute lung injury, which is the leading cause of death in COVID-19 patients with ARDS [46]. Acute lung injury is a hypoxic respiratory insufficiency caused by noncardiogenic pathogenic factors and may develop into ARDS in severe cases [23]. The exogenous rhuACE2 attenuates acute lung failure in ACE2-knockout and wild-type mice [45].

Huang et al. observed upper respiratory tract symptoms such as rhinorrhea, sneezing, and sore throat in patients with SARS-CoV-2. In addition, all of the 41 patients involved in the study had pneumonia, and the common complication was ARDS (12 of 41 patients). The chest CT scan images of 40 patients showed a bilateral involvement with multiple lobules and areas of consolidation. In four patients, invasive mechanical ventilation was required; two of them had refractory hypoxemia and received extracorporeal membrane oxygenation [47]. Severe hypoxemia in compliant lungs may be related to a poor ventilation–perfusion ratio as a result of a hypoxic vasoconstriction reaction in severe COVID-19 instances that require mechanical ventilation. This mechanism is to blame for the significant mortality rates seen in COVID-19 patients, with up to 80% of those who required mechanical breathing dying [10].

#### 2.2.3. AT1R and the Cardiovascular System

Many COVID-19 patients presented with one or more cardiovascular disorders or developed cardiac damage later in their infection period, making cardiovascular diseases account for the majority of the severe cases [46].

Disseminated intravascular coagulation and pulmonary embolisms were detected in COVID-19 patients [48]. In addition, increased D-dimer and fibrinogen levels in COVID-19 patients indicate thrombotic formation, and the endothelial damage produced by SARS-CoV-2 infection enhances the coagulation process, resulting in the creation of microthrombi. These can travel through blood vessels to different internal organs, resulting in pulmonary embolisms, in addition to heart, kidney, and liver ischemic injuries. Moreover, frequent activation of the coagulation process referred to as COVID-19 coagulopathy leads to poor outcomes and high mortality rates [49,50,51,52]. The elevation of proinflammatory cytokines and chemokines during COVID-19 infection inhibits anticoagulation pathways, consequently promoting thrombin formation [53]. Recently, high levels of Ang II were detected in COVID-19 patients, possibly contributing to the thrombosis seen in these patients [46,54].

In the cardiovascular system (CV) system, AT1R overactivity has been associated with the development of several pathological conditions, including hypertension, vascular inflammation, atherosclerosis, and heart failure [55,56]. Furthermore, Ang II–AT1R binding activates many cascades in the vasculature, such as protein tyrosine phosphatases, NADPH oxidase, MAPK, and NO synthase, resulting in the expression of proinflammatory mediators, contraction, and endothelial dysfunction [57,58]. When AT1R is stimulated, NF-κB is activated, causing resident cells and endothelial cells to upregulate proinflammatory and procoagulant factors. Furthermore, the ACE–Ang-II–AT1R axis promotes epithelial to mesenchymal transformation, which increases proinflammatory factors while lowering endothelial permeability [56]. On the other hand, ACE2 counterbalances Ang II–AT1R effects by either stimulating an alternative pathway for Ang I degradation to produce Ang-(1–9), or by inactivating Ang II and hydrolyzing it to a heptapeptide Ang-(1–7). The latter stimulates vasodilation and activates anti-inflammatory, antifibrotic, and antithrombotic cascades via the MasR axis [59,60], as well as the protection of endothelial cell activity [57,58].

Taken together, overstimulation and imbalance of the RAS can be associated with CV responses observed in COVID-19 patients, including hypertension, inflammation atherosclerosis, thrombosis, and myocarditis. Both the CV and systemic ACE/ACE2 ratio could be crucial in the evolution of COVID-19 individuals, favoring either the cardiotoxic ACE–Ang-II–AT1R axis or the cardioprotective ACE2/Ang-(1–7)/MasR pathway [61]. Therefore, an adequate equilibrium of these axes in specific tissues and along the different phases of COVID-19 could reduce the severity of disease and mortality in affected people [56].

ARBs effectively block AT1R, antagonizing the main Ang-II actions and exhibiting protective pleiotropic effects against hypertension and CV inflammation, fibrosis, and thrombosis [62]. In COVID-19 patients with underlying CV illness and ACE inhibitors (ACEi)/ARB medication, it was suggested that elevated ACE2 would encourage SARS-CoV-2 infection and replication [63]. However, the European Society of Cardiology and the American Heart Association advised against stopping these maintenance treatments in COVID-19, especially when hypertension or heart failure were present [64]. ACE2 internalization could be dependent on heterodimerization with AT1R (after Ang-II binding); thus, both ACEi and ARB may prevent ACE2-mediated viral internalization and consequent propagation [65]. In contrast, some studies have indicated that ACE2 internalization can occur independently of AT1R via viral endosomal entry by the protease cathepsin-L [66]. Indeed, the recombinant receptor-binding domain (RBD) of the SARS-CoV-2 S-protein was also shown to induce ACE2 internalization [67]. In fact, ACE2 internalization was increased by Ang II and reduced by telmisartan in cells coexpressing AT1R [67]. Therefore, targeting this entry route may also decrease viral internalization. In addition, it has been shown that AT1R could interact with other GPCRs, such as bradykinin II receptors, which also influence ACE2 availability [67]. Finally, increasing evidence supports the advantageous effects of both therapies against CV damage in COVID-19. According to findings from more than 20 clinical trials, data indicate that ACEi and ARB decrease the viral load, avoid peripheral T cell depletion, and reduce plasma IL-6, CRP, and procalcitonin levels [68,69,70]. ACEi and ARB could reduce the severity of COVID-19, mainly in the hyperinflammatory phase or in subjects with previous CV failures [62].

#### 2.2.4. AT1R and the Nervous System

Regarding the complications of SARS-CoV-2, it is important to consider its effects on the brain. Loss of taste and smell was largely seen among COVID-19 patients. In fact, taste bud cells have an average life span of 10 days and renew continuously from a population of stem cells in the oral epithelium. Proinflammatory cytokines such as TNF-α, IFN-γ, and IL-6 activated through the AT1R, can impede stem cell proliferation, and may also reduce the lifespan of mature taste bud cells. Therefore, these proinflammatory cytokines may lead to dysgeusia. Anosmia may be caused by direct infection and disruption of the olfactory nerve by COVID-19. As a result, it is believed that infection of support cells and regenerative stem cells, rather than direct impacts on the olfactory nerve, may cause anosmia. The loss of supporting cells around olfactory neurons impairs their ability to function [71].

The consequences of SARS-CoV-2 on CNS have been studied, and it has been discovered that this virus produces complications such as seizures and cerebral hemorrhage [72]. Increased entry of inflammatory cytokines into the brain, as a result of the SARS-CoV-2, generated cytokine storm, which has been considered as a possible underlying mechanism explaining these problems [73].

In this context, Ang II could be an important element in determining the link between SARS-CoV-2 and the brain difficulties it causes. After SARS-CoV-2 binds to ACE2, the virus causes endocytosis of this receptor [74].

Low levels of ACE2 in the cells’ surface increase Ang II levels, as reported in patients with COVID-19 [75]. Normally, Ang II cannot cross the blood–brain barrier (BBB), but in conditions such as hypertension, high levels of Ang II lead to the BBB disruption. It has been demonstrated that Ang II increases the BBB permeability through activation of the AT1R [76]. Increased BBB permeability by Ang II can be a reason for the virus entry into the brain [77].

Moreover, this condition can explain SARS-CoV-2-induced cerebral hemorrhage in COVID-19 patients, as the association between early BBB breakdown and hemorrhage has been reported in patients treated with thrombolytic drugs [78]. However, after SARS-CoV-2 enters the brain [77], it may trigger endocytosis of ACE2, resulting in an increase in the levels of cerebral Ang II, which can cause the production of inflammatory cytokines by binding to its receptors on astrocytes [79].

On the other hand, increased levels of Ang II may contribute to neuronal loss in different parts of the brain. It has been revealed that Ang II causes dopaminergic neurons death and losartan, an AT1R antagonist, protects these neurons against apoptosis [80].

Ang II–AT1R binding contributes to increased sympathetic tone [81]. AT1R is localized at circumventricular organ sites and other cardinal integrative regulatory centers of the brain such as the hypothalamus and medulla oblongata [82]. Circulating Ang II can mediate activation of the sympathetic nervous system (SNS) by acting on circumventricular organs and the carotid body. This system is triggered during blood loss, hyponatremia, renal hypotension, general SNS activation, and infection. Notably, SNS overactivity is implicated in many comorbidities associated with mortality in COVID-19 [83]. Ang II interacts with the proinflammatory AT1R, and downstream products of these processes activate STAT3 and NF-κB transcription factors, which promote IL-6 production, and upregulates NF-κB and STAT3 with the production of other proinflammatory cytokines [84].

ACE2 overexpression in the hypothalamus has substantial antihypertensive and SNS dampening effects, raising Ang (1–7), compared with Ang II, and thus having therapeutic benefits in mice models of brain damage and stroke [85]. The effects of Ang II and Ang III are effectively mediated through ACE1/ACE2 receptor connections. Unfortunately, AT1R activity typically prevails in SARS-CoV-2 infection [86].

#### 2.2.5. AT1R and the Digestive System

The presence of SARS-CoV-2 RNA in stool samples of some patients and more frequently in those with diarrhea suggests that asymptomatic patients may have infectious viral particles in their stools, which could be a source of infection to others [10]. The digestive tract manifestations of SARS-CoV-2 patients include nausea, diarrhea, vomiting, lack of appetite, abdominal pain, and sometimes GI hemorrhage [38,39]. It has been shown that COVID-19 patients with GI tract manifestations were significantly more likely to require intensive care admission and progress to ARDS than those without GI symptoms (6.8% vs. 2.1%, *p* = 0.034, and 6.8% vs. 2.1%, *p* = 0.034, respectively) [87].

The GI RAS regulates a number of physiological functions in the intestine, including electrolyte homeostasis, digestion, peptide transport, glucose, salt, and water absorption, GI motility, and secretion through the intestinal epithelium [88,89,90]. The GI RAS effects depend mainly on the balance between ACE1 and AT1R [91]. AT1R is localized at the epithelial brush border and is highly expressed in the circular and longitudinal muscle layers and the myenteric plexus [92,93,94]. Furthermore, because ACE2 is highly expressed on the luminal surface of the GI tract [10,95], SARS-CoV-2 has been reported to colonize the GI tract, particularly the intestines. SARS-CoV particles have been identified only in epithelial cells of the intestinal mucosa but not in the esophagus and stomach [96,97].

Generally, the activation of the ACE1–AT1R axis induces vasoconstriction and is involved in the induction of apoptosis, vascular remodeling, atherosclerosis, and inflammation [98,99]. Moreover, ACE2 downregulation in the intestine is related to the hyperactivation of the classical RAS axis [10]. ACE2 is normally involved in the control of dietary amino acid homeostasis, antimicrobial peptide expression, and gut microbial ecology [95].

Gut dysbiosis is an important issue that should also be taken into consideration in SARS-CoV-2 infection due to its potential link in the disease progression. This pathology is found in several COVID-19 risk groups such as diabetic patients, elderly, and immunocompromised patients [10].

It has been shown that the colonic RAS induces colonic inflammation by stimulating TH17 activation. In fact, within the colonic mucosa, Ang II induces colitis via AT1R via the JAK2/STAT1/3 pathway. By increasing the release of TH17-polarizing cytokines (such as Il-6 and TGF-1) from colonic epithelial cells via the JAK/STAT pathway, Ang II promotes TH17 polarization both directly and indirectly [36]. When SARS-CoV-2 downregulates luminal ACE2 in the gut, the Ang II level is increased. More Ang II (and therefore less Ang 1-7) results in luminal AT1R activation and enhanced permeability, leading to the so-called leaky gut syndrome [100].

Moreover, SARS-CoV-2 infection may lead to degeneration of the gut–blood barrier, driving to systemic propagation of bacteria and endotoxins and resulting in a septic shock [10]. Interestingly, the gut microbiota diversity may have an important role in determining the course of this disease [10]. In this regard, a pilot study including 15 patients with COVID-19 found persistent alterations in the fecal microbiome during the time of hospitalization [101].

Finally, SARS-CoV-2 may infect hepatocytes, causing an increased inflammatory response, as well as thromboembolic events that culminate in liver damage [10]. In fact, together with IL-1 and TNF-α, the presence of Il-6 in the liver is a key inducer of various acute-phase proteins, including CRP [2]. On the other hand, independently of IL-6, Ang II induces CRP expression in the liver through activation of AT1R and specifically the ROS–MAPK–NF-κB pathway [2]. Additionally, aminotransferases and bilirubin levels are higher in COVID-19 patients at the time of admission to the hospital, which has also been connected to disease severity and progression to critical illness [38].

#### 2.2.6. AT1R and the Renal System

The juxtaglomerular (JG) cells, afferent arterioles of the kidney, contain prorenin. While the latter is released constitutively in its inactive form, activation of JG cells causes the cleavage of prorenin to renin. Once these cells are active, blood pressure drops, beta-activation occurs, and *macula densa* cells become activated in response to a lower sodium load in the distal convoluted tubule [102,103].

In COVID-19 patients, kidney function abnormalities were observed, including hematuria (26.7%) and proteinuria (43.9%) [104]. Acidosis and hyperkalemia could also be seen in patients with COVID-19 [38]. However, a frequent COVID-19 complication in the renal system is the prevalence of acute kidney injury (AKI) [38,39]. Two possible mechanisms that underlie AKI have been hypothesized—namely, the direct cytotoxic effects of SARS-CoV-2 and indirect cytokine-mediated damage [104]. In fact, the cytokine storm might exert hypoxia secondary due to respiratory failure, septic, and cardiogenic shock [39].

An evaluation of 12 autopsy cases revealed that 6 patients had viremia, and viral RNA was also detected in kidney tissues at concentrations exceeding viremia. As ACE2 is expressed in epithelial basal cells, glomerular parietal epithelial cells, and proximal tubules of the kidney, these findings show that SARS-CoV-2 travels through the bloodstream and directly injures the kidneys [105,106].

Increased activation of the RAS has been postulated to play a central role in the progression of chronic kidney disease (CKD) [107]. In fact, the Ang II–AT1R axis induces renal fibrosis and inflammation, which appear to contribute to CKD, as well as the development of end-stage kidney disease [18]. Several pathways are activated in renal fibrosis: (1) AT1R induction of p38-MAPK and JNK signaling induces TGF-β activation, which contributes to an epithelial–mesenchymal cell transition [108]; (2) EGFR activation by the Ang II–AT1R axis also activates TGF-β, which leads to renal fibrosis [109]; (3) Ang II–AT1R via NF-κB pathway induces various inflammatory effectors (IL-6, TNF-α) that affect inflammatory responses [108]. In fact, IL-6 activation of the renal JAK/STAT pathway contributes to Ang II-induced hypertension [110].

Finally, it is noteworthy to mention that AT1R polymorphisms may influence the activity of Ang II detected in the kidneys. Healthy Caucasians carrying the A1166 C-allele (AC or CC) polymorphism showed an inferior basal glomerular filtration rate (GFR) and basal renal plasma flow and had increases in GFR following treatment with the AT1R blocker treatment [111].

## 3. AT1R Blockers (ARBs): A Potential Treatment Strategy in COVID-19?

ARBs, a well-known antihypertensive drug group that blocks AT1R, have been postulated as tentative pharmacological agents to treat COVID-19-induced lung inflammation [112]. Therefore, it is common for individuals with hypertension to use ARBs in order to control their blood pressure. As its name suggests, this type of medication works by blocking the Ang II receptor, given its role in promoting the constriction of blood vessels and in increasing blood pressure [113,114]. Since these receptors are found in the heart, blood vessels, kidneys, and intestine [115,116], blocking their action helps in lowering blood pressure and preventing kidney and heart damage [117]. In addition, blocking AT1R may reduce acute lung injury and the response of inflammatory mediators [62].

There is a debate on the use of RAS inhibitors such as ARBs and ACEi in SARS-CoV-2 infection [118]. The role of these drugs in SARS-CoV-2 infection is unclear (Table 1).

Some researchers speculated that RAS inhibitors would contribute to a higher susceptibility and severity of SARS-CoV-2 infection due to observations that elderly patients with CVD, for whom ARBs and ACEi are routinely prescribed, were at a higher risk for more severe SARS-CoV-2 infection [63,127]. Moreover, selected experimental studies suggested that RAS inhibitors may increase ACE2 expression [128,129,130]. On the other hand, many other studies contradicted such claims and implied that RAS inhibitors are effective in SARS-CoV-2 patients. ARBs are then believed to reduce acute lung injury in SARS-CoV-2 viral illness by the following mechanisms [112,131,132,133,134,135]: (1) blockade of AT1R may reduce the detrimental effects of Ang-II; (2) administration of ARBs may increase ACE2 expression, which may reduce the detrimental effects of Ang-II [118]; (3) AT1R blockade at the cell surface may reduce the internalization of the virus and consequently limit the decrease in ACE2 caused by the infection [65,118].

Soleimani et al. investigated the impact of ARBs on hospitalized hypertensive patients (48%) and found that they did not worsen clinical outcomes after COVID-19 infection [119]. Additionally, Meng et al. showed that ARB administration for 1 year before COVID-19 infection improved clinical outcomes of COVID-19 patients with hypertension (40.5%) [70]. Previous administration of ARBs had no association with the number of days alive and out of the hospital in mild COVID-19 patients (50.6%) [120]. Imai et al. used particular ARBs to treat acid-induced acute lung damage in ACE2 mutant mice. The pharmacological suppression of AT1R was found to reduce the severity of lung injury [45]. Gabriel et al. reviewed the effect of different ARBs on ACE2 and AT1R expression and investigated whether treatment of permissive ACE2+/AT1R+ Vero E6 cells with ARBs alters SARS-CoV-2 replication in vitro in an Ang II-free system. Azilsartan, eprosartan, irbesartan, and telmisartan induced a higher expression of ACE2 in Vero E6 cells than in untreated cells, which also have increased expression of AT1R, the binding of which to these blocking molecules was reduced. Interestingly, there are significant differences in the ability of ARBs to induce the overexpression of ACE2 once bound to AT1R, since some drugs at 7 µM (e.g., telmisartan) led to higher ACE2 increase than others when used at higher concentration (e.g., irbesartan at 60 µM) [117]. It is also known that following Ang II binding, AT1R triggers ACE2 cleavage and shedding, dependent on the p38–MAPK pathway, resulting in reduced cell surface expression [136,137]. In this regard, azilsartan, candesartan, losartan, and telmisartan have previously been demonstrated to have opposing effects on the MAPK pathway, which would preclude ACE2 excision [138,139].

The production of SARS-CoV-2 particles in supernatant of ARB-pretreated infected Vero E6 cells was evaluated at 24, 48, and 72 h.p.i. An increase in viral RNA expression in 24 h.p.i was significant for three of the ARBs used—azilsartan, eprosartan, and irbesartan. Interestingly, these compounds were likewise effective, positive modulators of the ACE2 protein expression. Moreover, the expression of viral proteins was evaluated inside of cells infected with SARS-CoV-2 after treatment with ARBs. These same three compounds were shown to induce the expression of S-protein, thus confirming the evidence that these compounds increase the multiplication of SARS-CoV-2. Results indicated that Vero E6 cells previously treated with ARBs for 72 h exhibit relative increases in ACE2 expression and SARS-CoV-2 production. Interestingly, scientists commented that Vero E6 cells were treated with ARBs without adding Ang II, which has been reported as a biomarker of severe COVID-19 [117]. In addition, increased SARS-CoV-2 production found in Vero E6 cells was not observed in the preliminary investigations recently performed on two human cell lines—namely, Caco-2 (intestinal epithelia origin) and Calu-3 (lung epithelia origin). In this same study, it was suggested that one of the possible reasons for this is the low expression of AT1R in human lung tissues, compared with that in other organs such as the kidneys [117,140]. Scientists suggested that a broader spectrum of human cell lines must be examined to see if an acceptable human cellular model could be found in which ARBs could upregulate ACE2 expression and SARS-CoV-2 production [117].

Conversano et al. sought to describe patient characteristics, ongoing pharmacological treatment at the time of admission, and any link between chronic usage of ACE inhibitors/ARBs and negative COVID-19 clinical outcomes in 191 patients. Data demonstrated that chronic treatment with ACE inhibitors/ARBs did not result in increased mortality, worse clinical presentation, or impairment of renal function in pneumonia patients who were hospitalized. Overall mortality was significant (22%), and age and comorbidities such as heart failure and chronic kidney disease were found as indicators of poor prognosis, all of which were consistent with previous reports from Wuhan, China [141].

Data from retrospective studies from COVID-19 patients have provided some evidence to support that hypothesis [142,143,144,145,146]. Rothlin et al. conducted a pharmacological analysis that suggested telmisartan as the best candidate to study [147,148]. A multicenter clinical trial was conducted on 162 patients from 18 years of age hospitalized with COVID-19 (80 in the standard care and 78 in the telmisartan-added-to-standard care group) to assess whether 80 mg of telmisartan given twice daily would be effective in reducing lung inflammation and CRP levels at 5 and 8 days of treatment in COVID-19 hospitalized patients. For primary outcomes (5 and 9 days), the baseline absolute CRP serum levels were 5.53 ± 6.19 mg/dL and 9.04 ± 7.69 mg/dL in the standard care and telmisartan-added-to-standard care groups, respectively. On day 5, patients in the telmisartan-added-to-standard care group had a lower absolute CRP serum level than patients in the standard care group (standard care 6.06 ± 6.95 mg/dL; telmisartan-added-to-standard care 3.83 ± 5.08 mg/dL). Additionally, CRP serum levels were lower at day 8 in patients treated with telmisartan than those in the standard care group (control: 6.30 ± 8.19 mg/dL; telmisartan: 2.37 ± 3.47 mg/dL). Considering that baseline CRP was higher in the telmisartan arm, treatment with the AT1R antagonist resulted in an inversion of CRP at days 5 and 8. In this study, the differences observed in CRP plasma levels between telmisartan and control groups suggest an anti-inflammatory effect of ARB [149]. This effect may have been clinically relevant considering that patients with high CRP levels are more likely to have severe complications [149,150]. For secondary outcomes, patients who received telmisartan in addition to standard care had a median discharge time of 9 days, compared with 15 days in those who received only standard care [149]. However, no conclusive data from a prospective randomized trial on the use of ARBs on COVID-19 patients are available [147].

Interestingly, the PRAETORIAN-COVID trial is a double-blind, placebo-controlled 1:1 randomized trial to assess the effect of ARB valsartan, compared with placebo on the occurrence of ICU admission, mechanical ventilation, and death within 14 days of randomization in hospitalized SARS-CoV-2–infected patients (*n* = 651). This design trial will provide valuable insights into the use of ARB in SARS-CoV-2 infected patients and may contribute to improved treatment recommendations for a large group of patients in this time of the global COVID-19 pandemic [118].

It is also interesting to point out the action of other drugs rather than ARBs in inactivating the RAS signaling such as NSAIDs or bioactive proteins. In fact, Oh et al. have found that three NSAIDs (indomethacin, 6-methoxy-2-naphthylacetic acid, rofecoxib) showed high levels of binding affinity with RAS proteins, which could subsequently block RAS and its damaging effects induced by SARS-CoV-2 [151]. Additionally, bioactive proteins play critical functions in inhibiting the RAS signaling pathway. Some of them were isolated from the leaves of *Morus alba* and showed great potential in fostering anti-gout arthritis by blocking the RAS signaling pathway [152].

## 4. Conclusions

Two years have passed since the outbreak of SARS-CoV-2 and its variations, which have posed a threat to human health around the world. The virus, which targets RAS, dysregulates it, and causes hyperactivation of its classical deleterious axis (Ang II–AT1R), characterized mainly by vasoconstriction, hyperinflammation, oxidative stress, and cell proliferation, as well as proapoptotic, proangiogenic, prothrombotic, prohypertrophic, and profibrotic events. In reality, Ang II, a critical constituent of RAS, primarily coordinates its endocrine, autocrine, paracrine, and intracrine effects through AT1R, a GPCR that stimulates a number of signaling pathways, including MAPK, NF-B, ROS, tyrosine kinase, etc. This leads to a variety of disorders in many organs and tissues, such as the lungs, heart, gonads, skin, and brain. Finally, several research studies conducted so far have argued that, like vitamin D, RAS inhibitors (especially ARBs) might be key elements to treat COVID-19, which affect renal, pulmonary, cardiovascular, immune, and brain functions.

## Figures and Tables

**Figure 1 molecules-27-02048-f001:**
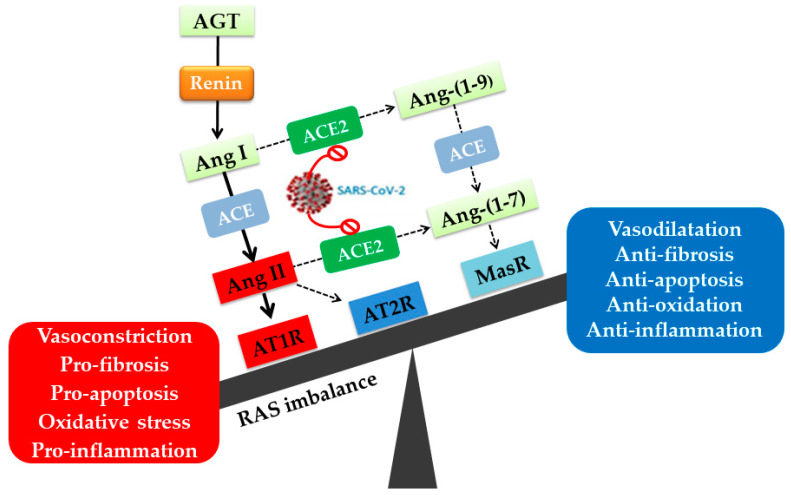
**The dysregulation of the renin-angiotensin system (RAS) caused by SARS-CoV-2 infection**. Upon SAR-CoV-2 entry, the downregulation of ACE2 decreases its ability to degrade Ang I and Ang II into Ang-(1-9) and Ang-(1–7), respectively. This is marked by excessive activation of the classical ACE-Ang II AT1R axis of RAS characterized by vasoconstriction, inflammation, apoptosis, fibrosis, and oxidative stress. AGT: angiotensinogen; Ang I: angiotensin; Ang II: angiotensin II; AT1R: angiotensin II type I receptor; AT2R: angiotensin II type II receptor; ACE2: angiotensin-converting enzyme 2; MasR: Mas receptor.

**Figure 2 molecules-27-02048-f002:**
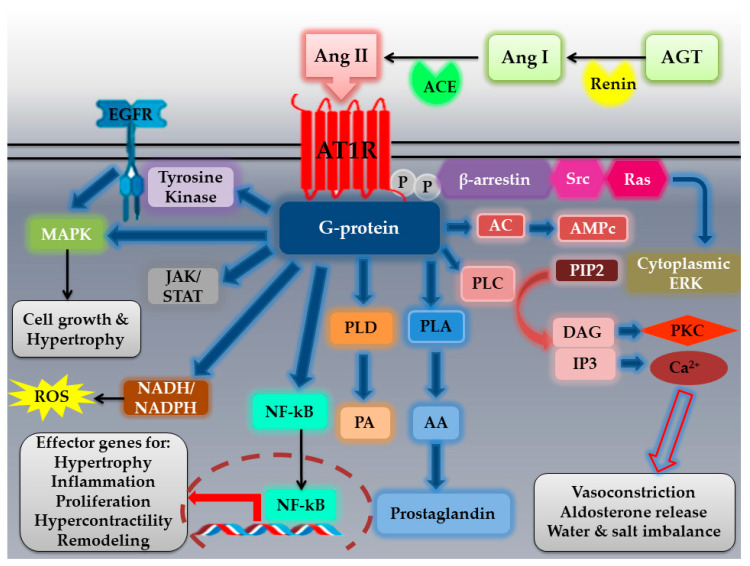
**Angiotensin II type I receptor (AT1R) activation by angiotensin II (Ang II) and its signaling pathways.** Ang II is produced from the conversion of AGT to Ang I by renin, and then, Ang I is cleaved by ACE into Ang II. Ang II is recognized by a G-protein-coupled receptor, AT1R. The agonist ligand (Ang II) binding to AT1R activates several signal transduction pathways, including noncanonical downstream effectors such as PLA, PLD, and PLC generating AA, PA, and IP3 and DAG, respectively. In turn, IP3 and DAG lead to Ca^2+^ and PKC activation, respectively implicated in water and salt balance, as well as aldosterone release and vasoconstriction. Canonical GPCR signaling activates receptor tyrosine kinases such as EGFR, as well as activating the NADPH complex, resulting in the generation of ROS, a potent second messenger implicated in oxidative stress. In addition, AT1R stimulation results in activation of MAPK and NF-κB transactivation. Together, this leads to pathophysiological responses such as hypertrophy, hypercontractility, proliferation, matrix production, inflammation, vascular remodeling, and hypertension. JAK/STAT activation by AT1R is involved in promoting cellular survival, migration, adhesion, and apoptosis. Moreover, AT1R rapidly undergoes desensitization through phosphorylation, which leads to β-arrestin recruitment, binding, and activation of MAPK-dependent signaling cascade. In fact, β-arrestin serves as a scaffold for signaling effectors such as Src, resulting in downstream activation of cytoplasmic ERK. AGT: angiotensinogen; Ang I: angiotensin I; ACE: angiotensin-converting enzyme; Ang II: angiotensin II; AT1R: angiotensin II type 1 receptor; EGFR: epidermal growth factor receptor; JAK/STAT: Janus kinase–signal transducer and activator of transcription; NADH: nicotinamide adenine dinucleotide hydrogen; NADPH: nicotinamide adenine dinucleotide phosphate; ROS: reactive oxygen species; NF-κB: nuclear factor κ-light-chain-enhancer of activated B cells; PLD: phospholipase D; PLA: phospholipase A; PLC: phospholipase C; PA: phosphatidic Acid; AA: arachidonic acid; PIP2: phosphatidylinositol 4,5-biphosphate; DAG: diacylglycerol; IP3: inositol 1,4,5-triphosphate; PKC: protein kinase C, Ca^2+^: calcium; AC: adenylate cyclase; AMPc: cyclic adenosine monophosphate; ERK: extracellular signal-regulated kinase.

**Figure 3 molecules-27-02048-f003:**
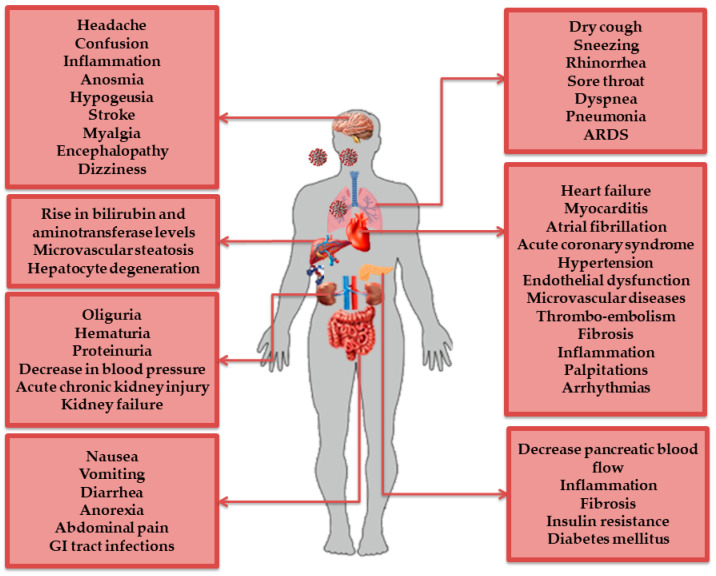
**COVID-19 associated multiorgan diseases.** ACE2 was found to be expressed in the lungs, the main entry route, as well as several tissues (including vessel endothelial and smooth muscle) and other vital organs (brain, heart, kidney, intestine, liver, etc.). Pulmonary and extrapulmonary manifestations are marked in SARS-CoV-2 infection and are summarized in this figure. Organ dysfunction includes both direct SARS-CoV-2 cytotoxic effects and cytokine-mediated damage.

**Figure 4 molecules-27-02048-f004:**
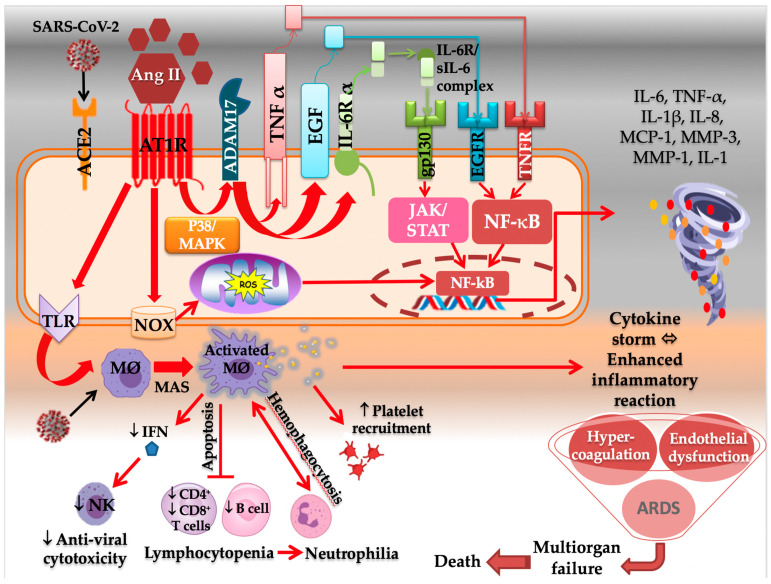
**The immunopathological features of SARS-CoV-2 infection.** SARS-CoV-2 binding to its cellular receptor (ACE2) favors its subsequent downregulation and the dysregulation of the RAS balance, which will result in a raise in Ang II level. The latter binds to AT1R and activates a series of signaling pathways implicated in the immunopathology process of the virus. Ang II increases ROS production through AT1R-dependent induction of NADPH oxidase (Nox), which will result in mitochondrial dysfunction and DNA damage, and will help the induction of inflammatory cytokines through NF-κB. ADAM17 is also activated by the Ang II–AT1R axis, via the intracellular p38/MAPK, and will digest the membrane forms of EGF and TNF-α generating their soluble forms, all of which also stimulate the NF-κB pathway through their specific receptors. The NF-κB pathway is involved in the pathogenesis of COVID-19, as well as in the induction of cytokine storm by inducing the expression of proinflammatory genes, including IL-6, chemokines, and growth factors. Moreover, ADAM17 cleaves the membrane-bound IL-6Rα to the soluble form (sIL-6Rα), which forms a complex with IL-6 and binds to gp130 and will result in the activation of JAK/STAT3 and the induction of excess IL-6. Following the activation of TLRs, macrophages are activated in large amounts, causing macrophages activation syndrome (MAS), which will stimulate the excess inflammatory cytokines secretion. The inflammatory response to SARS-CoV-2 also consists of lymphopenia manifested as a reduction in CD4+ and CD8+ T cells. Such impaired T-cell responses can result from deficient IFN production, which could result in the profound exhaustion of NK cells. These events lead to an imbalance in the innate/acquired immune response, delayed viral clearance, and unusual predominance of hyperstimulated macrophage and neutrophil in targeted injured tissues. The hemophagocytosis-like syndrome leads to ineffective viral cytotoxicity and weak antibody production. The uncontrolled amplification of cytokine production leads to endothelial dysfunction, ARDS, tissue damage, and multiorgan failure, which is the starting point of a progression towards the serious and fatal complications of COVID-19. AT1R: angiotensin type I receptor; SARS-CoV-2: severe acute respiratory syndrome coronavirus 2; Ang II: angiotensin II; TMPRSS2: transmembrane serine protease 2; ADAM17: a disintegrin and metalloprotease 17; MAPK: mitogen-activated protein kinase; TNFα: tumor necrosis factor α; EGF: epidermal growth factor, IL-6R α: interleukin 6 receptor α; sIl-6: soluble Interleukin 6, EGFR: epidermal growth factor receptor, TNFR: tumor necrosis factor receptor, gp130: glycoprotein 130; JAK/STAT: Janus kinase–signal transducer and activator of transcription; NF-κB: nuclear factor κ-light-chain-enhancer of activated B cells; ROS: reactive oxygen species; TLR: Toll-like receptor; NOX: NADPH oxidase; MØ: macrophages; IFN: interferon; NK: natural killer; MMP-1: matrix metalloproteinase -1; MMP-3: matrix metalloproteinase-3.

**Table 1 molecules-27-02048-t001:** Summary of clinical studies and trials associated with hypertension and Ang II type 1 receptor blocker (ARB) administration in relation to COVID-19 infection.

Antihypertensive Drugs Used (% of Population)	Period of ARB Intake before COVID-19 Infection	Main Outcomes	References
ARBs (48%)	≥7 days after hospital admission	ARB treatment did not worsen clinical outcomes during COVID-19 infection in hypertensive patients.	[119]
ARBs (50.6%)	Not mentioned	Previous administration of ARBs had no association with the number of days alive and out of the hospital in mild COVID-19.	[120]
ARBs (31.34%)	≥1 week before the onset of infection	ARB administration, before the onset of infection, significantly lowered the mortality rate in COVID-19 patients.	[121]
ARBs (49.3%)	Not mentioned	ARB administration had no effect on the severity and mortality of COVID-19.	[122]
ARBs (40.5%)	For >1 y	ARB administration improved clinical outcomes of COVID-19 patients with hypertension.	[70]
ARBs (31.8%)	Not mentioned	ARBs were not associated with the severity or mortality of COVID-19 patients with hypertension.	[123]
ARB (30.8%)	≥1 month before hospital admission	ARB administration reduced the risk of death during hospitalization in COVID-19 hypertensive patients.	[119]
ARBs (51%)	≥3 months before study conduction	ARB administration lowered the risk of hospitalization and intubation or death with COVID-19; long-term use of ARBs might decrease the risk of COVID-19 in hypertensive patients.	[124]
ARBs (29.7%)	Unknown	Administration of ARBs did not affect the chance of recovery in COVID-19 patients with hypertension or heart failure.	[125]
ARBs (17%)	Not mentioned	Previous use of ARBs reduced the risk of mortality in patients hospitalized with COVID-19 infection.	[126]

## Data Availability

Not applicable.

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
