# Peer review of "Angiotensin II Type I Receptor (AT1R): The Gate towards COVID-19-Associated Diseases"

_molecules, 2022, doi:10.3390/molecules27072048_

Round 1

Reviewer 1 Report

The topic is pretty relevant to the journal's readership and the author has also highlighted how this current research can impact the field. In general, I found the manuscript to be very well written and all the sections were clear and concise and had a very good flow between paragraphs. The title is clear and representative of the study. The abstract summarizes the work done concisely. The Introduction provides adequate background information for the reader, and well-framed and supported by data and citations. The Conclusion adequately highlights the strengths of the study.

However, in order to improve the manuscript, author should include some more information regarding the RAS signaling pathway.

  1. Inhibition of proinflammatory stimuli of tissues and/or cells by inactivating the RAS signaling pathway was identified as the key anti-COVID-19 mechanism of NSAIDs. (https://doi.org/10.1038/s41598-021-88313-5)
  2. The promising mechanism of M. alba L. leaves against gout was connected to a hub mechanism against gout might be to inhibit anti-arthritis immunity in synoviocytes by blocking the RAS signaling pathway. (https://doi.org/10.3390/ijms22179372)

Author Response

Dear reviewer,

First of all, we would like express our sincere gratitude for your valuable time in reading and reviewing our manuscript.

Secondly, we would like to thank you for all your delightful comments and suggestions that hopefully we have responded to them successfully.

Responses to comments:

The topic is pretty relevant to the journal's readership and the author has also highlighted how this current research can impact the field. In general, I found the manuscript to be very well written and all the sections were clear and concise and had a very good flow between paragraphs. The title is clear and representative of the study. The abstract summarizes the work done concisely. The Introduction provides adequate background information for the reader, and well-framed and supported by data and citations. The Conclusion adequately highlights the strengths of the study.

However, in order to improve the manuscript, author should include some more information regarding the RAS signaling pathway.

  1. Inhibition of proinflammatory stimuli of tissues and/or cells by inactivating the RAS signaling pathway was identified as the key anti-COVID-19 mechanism of NSAIDs. (https://doi.org/10.1038/s41598-021-88313-5)

Done (Added in the last section of paragraph 3)

  1. The promising mechanism of M. alba L. leaves against gout was connected to a hub mechanism against gout might be to inhibit anti-arthritis immunity in synoviocytes by blocking the RAS signaling pathway. (https://doi.org/10.3390/ijms22179372)

Done (Added in the last section of paragraph 3)

Reviewer 2 Report

Review report on “The angiotensin II type I receptor: a gate towards COVID-19 associated-diseases”

In general, this is an original, up-to-date and very relevant review to the field that provides detailed information about the role of AT1R in the COVID-19 associated diseases. The value and originality of the review is that it tackles the effects of RAS system dysregulation caused by SARS-CoV-2 infection in several systems, not only in the respiratory and cardiovascular as commonly done. In this way, the authors fill a gap in knowledge and provide a global approach to the several COVID-19-associated conditions. The references are accurate and the majority are current. However, in some parts the article gets repetitive or loses the thread of the message. Overall, special attention should be paid to integrate the sections and avoid repetition or information overload. Some parts can be summarized. I suggest the following changes in order to improve the message of this review before publication and solve some of the mentioned issues:

  1. The sentence “The physiological effects of ACE and ACE2 are mediated by two Ang II receptors, type 1 (AT1R) and type 2 (AT2R) [12-14].” at the beginning of paragraph 4 in Introduction is confuse, please clarify.

  1. The addition of a figure with a diagram of the two regulatory axis of RAS (the damaging: ACE-AngII-AT1R and the protective ACE2-Ang 1-7-MasR) in the introduction section in physiological conditions, and the dysregulation caused by SARS-CoV-2 infection (pathological condition) would be helpful to present the topic of the review and clearly localize AT1R in the RAS system for non-familiarized readers.
  2. The basis of RAS dysregulation caused by SARS-CoV-2 infection is explained in the introductory section and repeated in many of the paragraphs later. Indeed, it is mentioned in each paragraph as if has not been explained before. Maybe it should just be presented and explained in the introductory section and then, on the other sections focus just on the effects of that dysregulation in the different systems.
  3. In the section 2.1.1 it would be interesting to mention something about the ligand-independent activity of AT1R in certain diseases. Although AT1R activation is triggered by Ang II in physiological conditions, an Ang II-independent activation of AT1R by mechanical stress or AT1R autoantibodies has been observed in hypertension, preeclampsia and aldosteronism amongst others. The following work provides information about it: 10.1111/bph.14213.
  4. In the section 2.1.2 a link to the general topic of the review is lacking. Note that all the bibliography refers to the association between the polymorphism and cardiovascular diseases. This section should be highly summarized or a relationship between the different polymorphisms and the susceptibility to SARS-CoV- 2 infection or severity of COVID-19 should be provided. A recent work about that is available (10.1016/j.meegid.2022.105227).
  5. The section 2.2.1 (AT1R and the immune system) is overloaded of information and it is difficult to recognize the main topic of each paragraph (mainly paragraphs 3, 4 and 5 of this section), it gets the reader out of focus. Information should be summarized and reorganized inside each paragraph to make the message clearer. The emphasis should be put on the intracellular pathways activated by Ang II-AT1R overstimulation and the damaging effects on the tissues mediated by inflammatory effectors.
  6. In the section 2.2.2 check the second paragraph for repeated ideas/phrases about the increase in vascular permeability.
  7. In the last paragraph of section 2.2.3 says “ACE2 internalization is dependent on heterodimerization with AT1R (after Ang-II binding), []”. This statement makes it appear that ACE2 needs AT1R for its internalization in all situations. However, in the presence of coronavirus surface components (S-protein, RBD domain of S protein or lentivirus pseudotyped with S-protein) many authors have demonstrated ACE2 internalization in the absence of AT1R in HEK293T cells (10.1016/j.virusres.2008.03.004, 10.1038/cr.2008.15; doi.org/10.1016/j.lfs.2021.120284). According to a recent paper (doi.org/10.1016/j.lfs.2021.120284), AT1R seems to modulate SARS-CoV-2 induced ACE2 internalization in a ligand dependent manner. Interestingly, the authors showed that in the presence of AT1R Ang II increases ACE2 internalization and the antagonist telmisartan decreases it. Given these evidences, it would be more accurate to say that ACE2 internalization is modulated by AT1R, not dependent of this receptor.

  1. In the section 2.2.3 the recent finding that Ang II is a potent pro-internalizing factor for ACE2 internalization induced by SARS-CoV-2 in the presence of AT1R could be mentioned (doi.org/10.1016/j.lfs.2021.120284). Also, it is worth mentioning the anti-internalizing action of telmisartan in this conditions.

Author Response

Dear reviewer,

First of all, we would like express our sincere gratitude for your valuable time in reading and reviewing our manuscript.

Secondly, we would like to thank you for all your delightful comments and suggestions that hopefully we have responded to them successfully.

Responses to comments:

Review report on “The angiotensin II type I receptor: a gate towards COVID-19 associated-diseases”

In general, this is an original, up-to-date and very relevant review to the field that provides detailed information about the role of AT1R in the COVID-19 associated diseases. The value and originality of the review is that it tackles the effects of RAS system dysregulation caused by SARS-CoV-2 infection in several systems, not only in the respiratory and cardiovascular as commonly done. In this way, the authors fill a gap in knowledge and provide a global approach to the several COVID-19-associated conditions. The references are accurate and the majority are current. However, in some parts the article gets repetitive or loses the thread of the message. Overall, special attention should be paid to integrate the sections and avoid repetition or information overload. Some parts can be summarized. I suggest the following changes in order to improve the message of this review before publication and solve some of the mentioned issues:

1. The sentence “The physiological effects of ACE and ACE2 are mediated by two Ang II receptors, type 1 (AT1R) and type 2 (AT2R) [12-14].” at the beginning of paragraph 4 in Introduction is confuse, please clarify.

A clarification was done.

2. The addition of a figure with a diagram of the two regulatory axis of RAS (the damaging: ACE-AngII-AT1R and the protective ACE2-Ang 1-7-MasR) in the introduction section in physiological conditions, and the dysregulation caused by SARS-CoV-2 infection (pathological condition) would be helpful to present the topic of the review and clearly localize AT1R in the RAS system for non-familiarized readers.

A figure was added.

3. The basis of RAS dysregulation caused by SARS-CoV-2 infection is explained in the introductory section and repeated in many of the paragraphs later. Indeed, it is mentioned in each paragraph as if has not been explained before. Maybe it should just be presented and explained in the introductory section and then, on the other sections focus just on the effects of that dysregulation in the different systems.

Done

4. In the section 2.1.1 it would be interesting to mention something about the ligand-independent activity of AT1R in certain diseases. Although AT1R activation is triggered by Ang II in physiological conditions, an Ang II-independent activation of AT1R by mechanical stress or AT1R autoantibodies has been observed in hypertension, preeclampsia and aldosteronism amongst others. The following work provides information about it: 10.1111/bph.14213.

Done (briefly added since it is not the main topic of the review)

5. In the section 2.1.2 a link to the general topic of the review is lacking. Note that all the bibliography refers to the association between the polymorphism and cardiovascular diseases. This section should be highly summarized or a relationship between the different polymorphisms and the susceptibility to SARS-CoV- 2 infection or severity of COVID-19 should be provided. A recent work about that is available (10.1016/j.meegid.2022.105227).

This section was summarized and a brief relationship between the polymorphisms and COVID-19 susceptibility was added

6. The section 2.2.1 (AT1R and the immune system) is overloaded of information and it is difficult to recognize the main topic of each paragraph (mainly paragraphs 3, 4 and 5 of this section), it gets the reader out of focus. Information should be summarized and reorganized inside each paragraph to make the message clearer. The emphasis should be put on the intracellular pathways activated by Ang II-AT1R overstimulation and the damaging effects on the tissues mediated by inflammatory effectors.

Done

7. In the section 2.2.2 check the second paragraph for repeated ideas/phrases about the increase in vascular permeability.

Done

8. In the last paragraph of section 2.2.3 says “ACE2 internalization is dependent on heterodimerization with AT1R (after Ang-II binding), []”. This statement makes it appear that ACE2 needs AT1R for its internalization in all situations. However, in the presence of coronavirus surface components (S-protein, RBD domain of S protein or lentivirus pseudotyped with S-protein) many authors have demonstrated ACE2 internalization in the absence of AT1R in HEK293T cells (10.1016/j.virusres.2008.03.004, 10.1038/cr.2008.15; doi.org/10.1016/j.lfs.2021.120284). According to a recent paper (doi.org/10.1016/j.lfs.2021.120284), AT1R seems to modulate SARS-CoV-2 induced ACE2 internalization in a ligand dependent manner. Interestingly, the authors showed that in the presence of AT1R Ang II increases ACE2 internalization and the antagonist telmisartan decreases it. Given these evidences, it would be more accurate to say that ACE2 internalization is modulated by AT1R, not dependent of this receptor. 

Done

9. In the section 2.2.3 the recent finding that Ang II is a potent pro-internalizing factor for ACE2 internalization induced by SARS-CoV-2 in the presence of AT1R could be mentioned (doi.org/10.1016/j.lfs.2021.120284). Also, it is worth mentioning the anti-internalizing action of telmisartan in this conditions.

Done
